

# Aerodynamic effects of leading edge erosion in wind farm flow modeling

Jens Visbech[1], Tuhfe Göçmen[1], Özge Sinem Özçakmak[1], Alexander Meyer Forsting[1],
Ásta Hannesdóttir[1], and Pierre-Elouan Réthoré[1]

[1]Department of Wind and Energy Systems, Technical University of Denmark (DTU), 4000 Roskilde, Denmark

**Correspondence:** Jens Visbech (jvima@dtu.dk)

**Abstract.** Leading edge erosion (LEE) can significantly impact the aerodynamic performance of wind turbines and thereby the overall efficiency of a wind farm. Typically, erosion is modeled for individual turbines where aerodynamic effects are only impacting the energy production through degraded power curves. For wind farms, the aerodynamic deficiency has the potential to also alter the wake dynamics, which will affect the overall energy production. The objective of this study is to
demonstrate this combined effect by coupling LEE damage prediction and aerodynamic loss modeling with steady-state wind farm flow modeling. The modeling workflow is used to simulate the effect of LEE on the Horns Rev I wind farm. Based on a 10-year simulation, the aerodynamic effect of LEE was found to be insignificant for the first few years of operation, but rapidly increases and reaches a maximum AEP loss of 2.9 % in the last year for a single turbine. When including the impact of LEE to the wakes behind eroded turbines, the AEP loss is seen to reduce to 2.7 % at the wind farm level.

**1   Introduction**

Erosion is often observed on wind turbine blades where the material of the leading edge has gradually been worn away over time. LEE may be caused by impacts of airborne particles such as rain droplets, sand, and hail, or by other factors such as UV radiation, strain from blade bending, or rapid temperature changes (Keegan et al., 2013). The impact of these factors on erosion varies from one location to another, but the most common damaging force is heavy rain occurring simultaneously with
high wind speeds. In Denmark and the UK, rain-induced LEE is a critical problem for many offshore wind farm operators, where in some instances blades have been repaired or changed after only a few years of operation (offshoreWIND, 2018). Compared with onshore turbines, offshore wind turbines operate more frequently at maximum tip speed due to higher average wind speeds. Further, offshore wind turbines are not affected by noise regulations that limit the maximum tip speed, allowing them to operate at greater tip speed (Herring et al., 2019), which benefits power production.

LEE has significantly impacted the wind energy industry in terms of repair costs. The erosion damages often require special kinds of repair solutions, such as the installation of protective shields or tapes, filling and injection coating, and resin injection for small surface cracks (Mishnaevsky, 2019). The cost of surface erosion repairs can vary depending on the extent of the damage, the location, and the size of the turbine. In a recent study, it was demonstrated that surface erosion is the largest contributor to unplanned repair costs (Mishnaevsky Jr. and Thomsen, 2020).



Damage prediction models, also referred to as lifetime prediction models, are used to estimate the damage state of the leading edge based on weather inputs such as wind speed and rain. They can be particularly useful for adequate planning and scheduling of maintenance actions. These models often rely on known or assumed materialistic fatigue strength properties obtained from rain erosion tests (RET) which are useful for predicting the erosion incubation period. Several studies have proposed damage models for predicting site-specific erosion damages (Visbech et al., 2023; Verma et al., 2021; Prieto and

Karlsson, 2021; Castorrini et al., 2021). However, the focus of these studies has solely been on structural defects.

Another important, but less documented cost related to LEE is the loss of aerodynamic efficiency. LEE on wind turbine blades roughens the surface, thereby causing aerodynamic performance deterioration. Airfoils used in shaping wind turbine blades are carefully designed to satisfy specific design requirements related to aerodynamic performance, geometric and structural reliability, etc. (Bak, 2022b). Even small perturbations to the surface geometry can significantly impact the desired airfoil

properties. The two main aerodynamic properties of an airfoil are its lift and drag coefficient. These normalized quantities describe the airfoil's ability to generate lift and drag and vary with angle-of-attack and Reynolds number. The lift-to-drag ratio is typically used as a proxy for aerodynamic efficiency, as it indicates how much undesired drag is required to generate a certain desired amount of lift. When an airfoil is exposed to LEE, the flow characteristics around it change. Several studies have investigated the effects based on high-fidelity methods such as computational fluid dynamics (CFD) (Li et al., 2010; Castorrini

et al., 2020; Wang et al., 2022; Meyer Forsting et al., 2022a), wind tunnel experiments (Bak et al., 2000; Kruse et al., 2021), or a combination of both (Maniaci et al., 2016; Kruse et al., 2018; Meyer Forsting et al., 2022b, 2023). Generally, LEE was found to cause a sharp and pronounced increase in drag. Additionally, a reduction in the pressure differential (between the pressure and suction side) leads to a reduction in lift.

2D airfoil properties can be used together with blade element momentum (BEM) theory to couple classical momentum

theory with the local forces acting on the blade sections (Glauert, 1935; Hansen, 2015). This allows for estimating the full rotor aerodynamics and thus blade forces, which, at the rotor level, are summarized by the power and thrust coefficients. Several studies have adopted this approach to quantify the effect of LEE on power production and AEP (Cappugi et al., 2021; Maniaci et al., 2020; Bech et al., 2018; Bak, 2022a) by replacing the baseline 2D airfoil performance with the one incorporating LEE. Still, it is not fully recognized by wind farm operators, that LEE affects energy production notably since it is extremely difficult

to validate from operational wind turbine data. This is due to the stochastic nature of the turbulent wind and the large year-to-year variability in the wind resources (Lee et al., 2020). However, a recent study by Panthi and Iungo (2023) investigated operational data from 53 GE 1.5 MW turbines from the Cedar Creek wind farm (US-CO), with the objective of quantifying AEP losses from LEE. Losses in the range of 3-8 % were observed from supervisory control and data acquisition (SCADA) data with the largest loss contributions coming from the low-wind speed operational regime.

As mentioned above, the main focus of former studies has been on the direct effect of LEE on energy production. A general reduction in aerodynamic efficiency will decrease a turbine's ability to convert kinetic energy into torque, but thus will also leave more energy for downstream rotors, as its wake deficit is diminished. For that reason, LEE effectively works as unintentional axial induction control, which is a well-known wake mitigation strategy. This added effect is only relevant in wind farms where wake effects play an important role, and could explain why it is commonly overlooked.



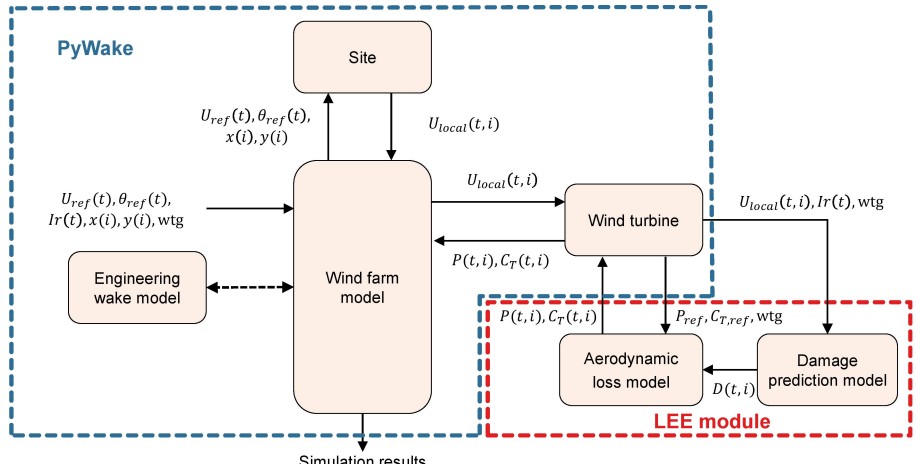

**Figure 1.** Overview of the modeling workflow.

Power and thrust coefficients are typically used in wind farm simulation tools such as PyWake (Pedersen et al., 2023) or FLORIS (NREL, 2021), and allow estimating wind farm energy production. These wind farm simulation tools rely on engineering wake models (Bastankhah and Porté-Agel, 2014; Jensen, 1983; Frandsen et al., 2006; Ott et al., 2011; Larsen et al., 2007) for estimating steady wind farm flow fields, which offer a balanced trade-off between prediction accuracy and computational costs.

In the present paper, we test the hypothesis that LEE directly affects wind farm energy production through degraded power and thrust curves, hence including its effect on wake losses. This is accomplished by modeling the temporospatial progression of erosion independently for each turbine within a wind farm and evaluating its effect on key aerodynamic properties influencing the power and thrust coefficients. This is achieved by coupling a damage prediction model with a fast aerodynamic LEE loss prediction tool and a steady wind farm flow model. We use this modeling framework to demonstrate how LEE-induced
power losses differ between an individual turbine and an entire wind farm through a case study. Finally, we use the probabilistic capabilities of the damage prediction model to propose a prioritized repair strategy based on Monte Carlo simulations.

The paper is structured as follows: Section 2 describes the overall modeling framework including a thorough description of the modules used for modeling wind farm aerodynamics under the influence of LEE. In section 4 and 5, the results obtained throughout the study are presented and discussed, respectively. Finally, the main conclusions are summarized in section 6.

## 2    Methodology

### 2.1    Modeling overview

The current section describes the methodology used for modeling the combined aerodynamic effects of LEE in wind farms. The overall workflow of the modeling framework is visualized in Figure 1. The framework revolves around a central wind farm



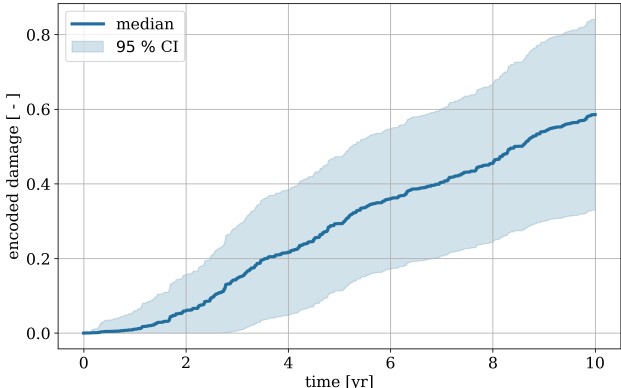

**Figure 2.** Example of the output from the damage prediction model based on 10 years of weather data as input. The graphs show the progression of the encoded damage level from a time series of half-hourly wind speed and rain data. The solid line represents the ensemble mean and the shaded area indicates the 95 % ensemble CI.

simulation tool that runs with an engineering wake model and is coupled to a LEE module. The LEE module consists of two
sub-modules; firstly, a damage prediction model is used to provide probabilistic damage estimates based on the site-specific time series of weather inputs and turbine operational data. The damage estimate is then passed on to an aerodynamic predictor which determines the blade-sectional aerodynamic losses. These sectional losses are combined to provide the final output in the form of eroded power and thrust curves. These properties are finally fed back to the central wind farm model and used in the computation of the wake deficits to update the wind farm flow field and turbine power production. The wind farm can be
simulated over a time series of wind speed, wind direction, and rain, to simulate the gradual development of erosion on each individual turbine in a wind farm. The damage state can be updated at each time step, or updated after a block of time steps to speed up the simulation time. It should also be mentioned, that since the modeling framework is modularized, it is very flexible and not limited to the setup used within this paper. The individual models can easily be substituted with other models, provided they take the same inputs/outputs.

**2.2 Damage prediction model**

The damage prediction model used in this study was originally proposed by Visbech et al. (2023) and the following description will only cover the model in relation to the scope of the present study. For detailed information, the authors refer to the original paper.

The model is based on an ensemble of 816 small feed-forward neural networks. The networks were trained with mesoscale
weather data and blade inspections from seven wind farms located on- and offshore in Northern Europe. The mesoscale weather data were provided as hourly time series of wind and precipitation and the blade inspections were obtained from a combination of manual, ground- and drone-based images. The purpose of the damage prediction model is to provide estimates of the erosion damage along the blade, based on time series of turbine-local wind speed and rain. Together with turbine-specific





operational characteristics, these are used to calculate rain impingement following industry recommendations (DNV, 2020),
which is the main predictor variable used by the model. The output from the damage prediction model is an encoded damage
value ranging between 0 and 1. The encoded damage from the model is directly related to a specific defect type and severity.
The categorization is based on the structural integrity of the blades and therefore represents the urgency for repair actions.
Also, the encoding scheme allows for continuous and realistic damage progression, similar to that observed from actual blade
inspections. Though unique to the blade inspections used for training the damage prediction model, the categorization scheme
is similar to others proposed in literature (e.g., Sareen et al., 2014; Gaudern, 2014).

Figure 2 demonstrates the output of the damage prediction model, based on a 10-years time series of wind speed and rain
used as input. The solid line represents the average encoded damage of a wind turbine. Since the model consists of an ensemble
of several hundred neural networks, it allows for making probabilistic damage estimates by incorporating uncertainty observed
from the blade inspections used during training. This is also visualized in Figure 2 by the 95% ensemble confidence interval
(CI) predicted by the damage model. Here, we observe the heteroscedastic uncertainty captured by the model, which can be
used to introduce realistic damage variability, similar to that observed in the field.

## 2.3 Aerodynamic loss categories

While the damage prediction model provides estimates of the structural erosion damage, it does not consider the associated
aerodynamic losses. Doing this requires information on the sectional degradation of the lift-to-drag ratio and maximum lift
coefficient due to damages along each blade. It is common to categorize blade inspection data into severity classes by judging
the risk of damage progression and potential repair costs, as done for the data underlying the damage prediction model.
Yet, severe structural damages, e.g., deep isolated cracks, do not lead to severe aerodynamic losses, whereas structurally
insignificant findings, like top coat erosion, do. Therefore an aerodynamic categorization of leading damages needs to be
performed independently from the structural assessment.

A standardized aerodynamic loss categorization scheme is yet to be established, as there is insufficient knowledge about
the detailed geometric realization of the damages encountered in the field and their corresponding frequency of appearance.
For aerodynamic loss assessment, the exact 3D damage topology needs to be known, as even features down to 50 $\mu$m can be
aerodynamically active. However, with the more frequent appearance of severe damages, a growing number of wind tunnel
and numerical investigations have been performed to quantify their aerodynamic impact (Sareen et al., 2014; Gaudern, 2014;
Ehrmann et al., 2017; Veraart, 2017; Meyer Forsting et al., 2022a, b, 2023). Applying similar damage topologies resulted in
comparable relative changes in the lift-to-drag ratio, despite the use of different airfoils. It is likely that this is also related
to having investigated similar Reynolds numbers ($\leq 6$ million) and thin airfoils ($\leq 21$ %) that are usually used in the outer
blade region. Aerodynamically, this is rooted in the impact of surface perturbations, also those caused by LEE, being related
to the ratio of surface feature height to boundary layer thickness. In turn, this is a function of the Reynolds number, whilst the
airfoil thickness is a proxy for the surface pressure gradient, which again influences the boundary layer. Generally, the biggest
drop in performance arises from roughness inducing premature boundary layer transition right at the leading edge, with the
aforementioned studies reporting losses between $35 - 50$ % with respect to the clean, free-transition baseline. Indeed, the





maximum loss registered for erosion-type damages never exceeded 64% and once roughness caused the transition to occur at the leading edge, the additional loss from more severe erosion was between $10 - 15\%$. As all available loss data was compiled for Reynolds numbers below 6 million, it is difficult to extrapolate these loss factors directly to modern wind turbines. Some blade regions could be operating at 15 million and as the relative losses are diminishing with increasing Reynolds number, the transition location moves gradually towards the leading edge. To assess the aerodynamic losses at higher Reynolds numbers 2D CFD computations were performed for four airfoils with relative thickness below 21 % (NACA63-418, FFA-W3-211, DU96-W-180, Risø-B1-18, Risø-C2-18) for a Reynolds number range from $5 - 15$ million and different levels of erosion. The latter was generated using the spectral approach detailed in (Meyer Forsting et al., 2022a) and combined with a forward-facing step (height of $1.5 \times 10^{-3}$ in chord units) on the suction side to represent the worst erosion level. The surface perturbations were directly resolved in those simulations. The entire workflow from surface grid generation to post-processing was automatized within the Python tool PyE2Dpolar for which details are given in (Meyer Forsting et al., 2022a, 2023).

The relative drop in lift-to-drag ratio diminishes by about 15 percentage points when going from 5 to 15 million in Reynolds number. In line with structural damage categorization and with the aim to create distinct aerodynamic loss categories, the losses were divided into 5 categories and labeled with letters to clearly distinguish them from structural classifications. The categories can be found in Figure 3.b. The first two categories capture the losses associated with the gradual movement of the transition location towards the leading edge over the entire blade section in question. At level $c$ transition occurs at the leading edge. Here it is assumed that transition is caused by pits and gouges. Additional losses are caused after causing early transition from surface roughness and sharp steps, as those forming between eroded and non-eroded areas. As the clean performance obtained in wind tunnels or simulations is unlikely to be matched in the field due to dust and other small-scale surface imperfections, the baseline airfoil performance to which the categories are indexed is a 60:40 mix of transitional and fully turbulent performance (Bak). Losses are provided for both, 5 and 15 million in Reynolds number and are representative of the performance loss reported in previous studies and those conducted as part of this paper.

## 2.4 Coupling aerodynamic and structural categories

The original modeling objective of the damage prediction model was to support site-specific repair and maintenance planning. This was done through an encoded damage score representing the damage state of a wind farm in relation to the urgency for repair. As previously mentioned, the purpose of this study is to investigate the aerodynamic effect of LEE for wind farm flow modeling, and for that reason, a relation between structural and aerodynamic defect categorization has to be established.

As mentioned in the previous section it is difficult to map aerodynamic to structural categories, however here this was attempted by matching the observable damage features. Figure 3.a shows the categorization of structural defects (defect type on vertical axis and defect severity on horizontal axis) used by the damage prediction model, and the corresponding encoded damage value. As mentioned, Figure 3.b shows the categorization of the aerodynamic losses (Reynold's number on vertical axis and categories on horizontal axis), obtained from the CFD simulations and literature review. Figure 3.c shows the final relation between between the encoded damage and the percentage lift-to-drag ratio. It is assumed that aerodynamic losses will grow quickly with the onset of structural damage accumulation as it causes the transition point to rapidly move towards the



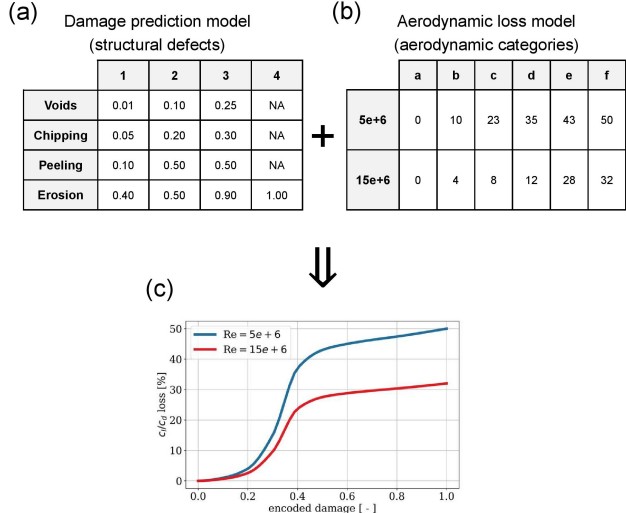

**Figure 3.** Mapping between aerodynamic loss and encoded structural defects. (a) shows the structural categories and associated encoded damage level, (b) shows the aerodynamic categories and associated lift-to-drag loss and (c) shows final coupling between the encoded damage level and aerodynamic loss.

leading edge. From then on the aerodynamic losses rise more slowly to reach its maximum once the leading edge is fully eroded.

## 2.5   Aerodynamic loss model

The main purpose of the aerodynamic loss model is to obtain the loss in annual energy production of a turbine due to leading-edge erosion and to obtain the power and thrust curves of turbines subjected to LEE that can be used by the wind farm simulation tool. In this paper, we use an adoption of the aerodynamic loss model introduced by Bak (2022a). The tool is a simplified blade-element momentum (BEM) theory model where the blade is divided into annular elements and the local losses are calculated at each annular element. The model is very light in its implementation as it is independent of the inflow
angle which significantly speeds up the computation. In addition, it includes a simplified tip correction model that only depends on the tip speed ratio (TSR), the blade radius, and the number of blades. The aerodynamic loss model was validated by Bak (2022a) in comparison to a classic BEM model, and it was found that the local power and thrust coefficients along the rotor radius compare well. The input parameters that are needed for the tool to estimate the AEP loss for a single turbine are listed in Table 1. In this particular study, the AEP loss is calculated at the wind farm level. Therefore the site parameters are not
introduced here but in the wind farm model.

The main input parameters to the aerodynamic loss model is the sectional loss in lift-to-drag ratio which is provided by the damage prediction model using the structural-to-aerodynamic coupling previously described. Figure 4 illustrates an example of the loss distribution along the blade where the highest lift-to-drag loss is reached towards the tip and the erosion severity



**Table 1.** Input parameters for AEP loss prediction tool).

| Wind Turbine Properties |
| --- |
| Number of blades |
| Rated power |
| Max Tip Speed |
| Drive-train efficiency |
| Optimum TSR |

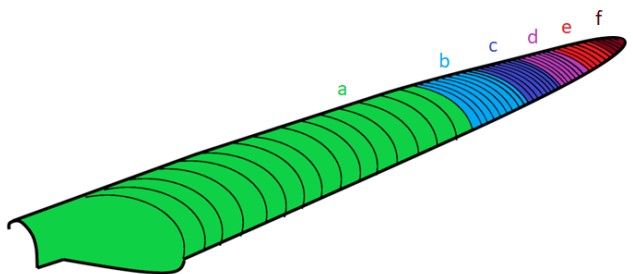

**Figure 4.** Blade sections and erosion securities introduced to the sections in terms of lift-to-drag losses. ('f' is the most severe erosion and 'a' is the clean case with no loss)

decreases towards the root. For this representation of the erosion distribution on the blade, it is assumed that the predicted

damage occurs at the tip of the blade and that it decreases towards the root, following a cubic relationship. Due to the fact that the most severe damage typically occurs towards the tip of the blade, and that the outer part of the blade plays a greater role in aerodynamics, a logarithmic discretization was chosen to divide the blade into sections. Figure  4 provides a visualization of this non-linear discretization.

In addition to the loss in lift-to-drag, there are more inputs that are not always available for a given turbine such as the

design lift coefficient and sectional lift-to-drag coefficient ratios for the clean blade. For those parameters, default values that are obtained from a test turbine are used.

An empirical relation for the thrust coefficient is used that depends on the tip speed ratio, and design lift coefficient. The tip loss and lift coefficient ratio of an eroded and clean reference case is included to find the thrust coefficient for the eroded case.

In the aerodynamic loss model, there are no specified control properties. It is assumed that the wind turbine is variable-speed

pitch regulated (VSPR) and operates at maximum power coefficient below rated wind speed. If subject to a constraint on tip speed which is violated before rated wind speed, a sub-region might occur, where the tips speed ratio is kept suboptimal.

Similarly, it is assumed that the wind turbine is capable of adequately shifting the pitch and rotational speed to adjust for the aerodynamic degradation of the lift and drag coefficients when exposed to erosion. LEE reduces the efficiency of the blade




which effectively shifts the rated wind speed to a higher value. As mentioned, these control properties are not specifically
defined but are assumed inherent for the wind turbine operation.

## 2.6    Wind farm flow modeling

In order to incorporate the long-term progression of LEE damage into wind farm response modeling, the steady-state behavior
of wake effects is of relevance to be included in the simulations. Steady-state engineering wake models offer a significant
advantage by enabling the computationally low-cost prediction of wind farm flow fields including turbine wakes, to assess
power capture and ultimately AEP. Compared to more detailed, often dynamic high-fidelity numerical tools such as CFD with
much higher computational cost, the engineering wake models have been shown to provide comparable accuracy, requiring
much lower complexity in their inputs in terms of flow properties and turbine characteristics (e.g., blade geometry, detailed
representation of the controller, etc.) (Göcmen et al., 2016).

Here in this study, the steady-state flow within the wind farm is represented using the open-source wind farm simulation tool
PyWake (version 2.5) (Pedersen et al., 2023). PyWake provides engineering models for estimating the wind farm flow fields,
including the Gaussian wake deficit formulation proposed by Bastankhah and Porté-Agel (2014) presented in equation 1 below.
It assumes that the wake spreads radially outwards, that the wake deficit follows a Gaussian shape and that the wake centerline
is aligned with the turbine rotor plane. It has shown very good agreement with field observations in several campaigns and
wake model benchmarks (e.g. Doubrawa et al., 2020), especially in the far wake region and it is utilized in this analysis to
estimate the flow characteristics behind eroded turbine(s).

$$
\frac{\Delta U}{U_\infty} = \left( 1 - \sqrt{1 - \frac{C_T}{8(k^* x/D + \epsilon)^2}} \right)
$$
$$
\times \exp \left( -\frac{1}{2(k^* x/D + \epsilon)^2} \left\{ \left(\frac{z - z_h}{D}\right)^2 + \left(\frac{y}{D}\right)^2 \right\} \right) \tag{1}
$$

where $\frac{\Delta U}{U_\infty}$ is the wake deficit normalized by the freestream velocity, $x, y, z$ are the streamwise, spanwise and vertical spatial
coordinates with $z_h$ as the hub height, $D$ is the turbine diameter, $C_T$ is the thrust coefficient of the turbine, and $k^*$ is the wake
expansion parameter defined by the local turbulence intensity (streamwise) at hub height $TI_h$ as (Niayifar and Porté-Agel,
2015):

$$k^* = 0.4 TI_h + 0.004 \quad \text{for} \quad 0.065 \leq TI_h \leq 0.15$$
$$k^* = 0.03 \qquad\qquad \text{for} \quad TI_h < 0.065 \quad \text{or} \quad TI_h > 0.15$$

Additionally, $\epsilon$ is proposed as a function of $C_T$ (Bastankhah and Porté-Agel, 2014), described by $\epsilon = 0.2\sqrt{\frac{1}{2}\frac{1+\sqrt{1-C_T}}{\sqrt{1-C_T}}}$.
Therefore, with the expected reduction in $C_T$ due to progression of LEE over time, the wake deficit described in equation 1 is
also anticipated to decrease.

The detailed specifications for the PyWake simulation setup can be found in Appendix A.



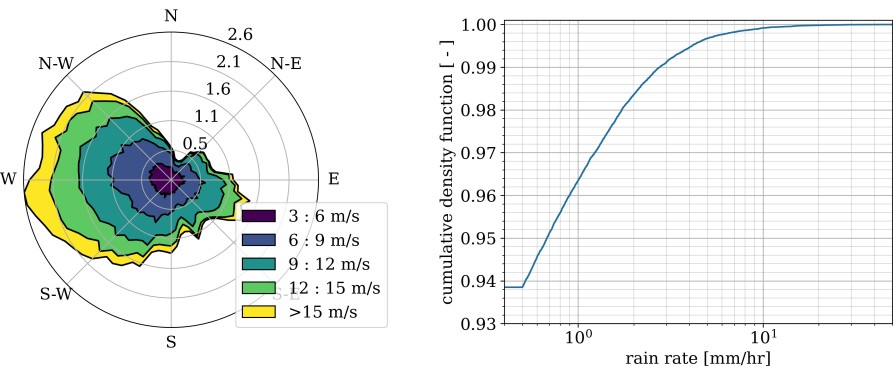

**Figure 5.** Weather conditions at Horns Rev I given by (left) a windrose with mean speed of 9.3 m/s and a mean wind direction of 252° and (right) the cumulative density function of rain with a mean annual rainfall of 1020 mm/yr. A lower bound threshold of 0.5 mm/hr is applied to rain data.

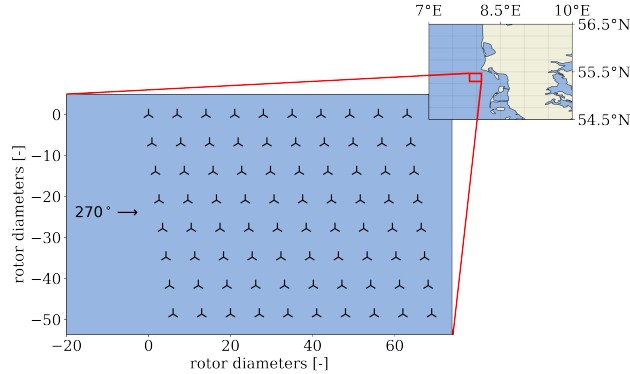

**Figure 6.** Layout and geographical location of Horns Rev I wind farm. The minimum spacing between turbines is 7 rotor diameters.

## 3   Case study: Horns Rev I

The modeling framework described in the previous section will be demonstrated through a case study for the Horns Rev I
offshore wind farm, which is located in the North Sea along the west coast of Denmark. The wind farm was commissioned in
2009 and consists of 80 Vestas V80-2MW wind turbines, yielding an installed capacity of 160 MW. The layout and geograph-
ical location of Horns Rev I are shown in Figure 6, where the minimum spacing between the turbines is 7 rotor diameters (560
m).

Weather data used in the study are obtained from the mesoscale weather research and forecast (WRF) model used in the New
235   European Wind Atlas (NEWA) (Witha et al., 2019). The data contain wind speed, wind direction, turbulent kinetic energy and
precipitation. The data are provided as time series with a 30-min time resolution and with a 3 km grid spacing. Precipitation



data are provided at surface level whereas wind data are provided at 75 m height and extrapolated to hub height using the power law with shear exponent $\alpha = 0.1$.

The wind climate at Horns Rev I is governed by westerlies coming from the sea, thereby providing a very good wind resource. The mean wind speed at hub height (70 m) is 9.3 m/s and the mean wind direction is 252°. The average turbulence intensity (TI) is 6.7 %. Figure 5 illustrates the statistical weather conditions at the site shown by a wind rose on the left and the exceedance probability of rain on the right. The annual rainfall was found to be 1053 mm/yr, with a large proportion falling in the autumn. The rain exceedance probability also shows that rain occurs around 6 % of the time. A lower bound threshold of 0.5 mm/hr is applied to the rain to account for the mesoscale model uncertainty, which is also observed from the sharp edge on the graph at this value.

## 4 Results and analysis

### 4.1 Deterministic simulations

The first step in the analysis is to compare the effect of eroded blades when modeling a single turbine against a full wind farm. To do this, we are running two simulations; one using a single Vestas V80 turbine, and one using the full Horns Rev I wind farm. For both simulations, 10 years of weather data and wind turbine specifications from Horns Rev I are used for simulating the operation of the wind turbines and their gradual aerodynamic degradation caused by LEE. For these simulations, only the ensemble mean from the damage prediction model is used as an output for updating the damage state of all the turbines, thereby providing a deterministic damage estimate only. In this case, the only damage variability comes from the local wind speed observed by each turbine. To account for randomness in the weather data, the 10 year simulations are run for 10 random seeds.

Figure 7 shows the estimated AEP loss of the eroded turbine and wind farm relative to its identical but non-eroded counterpart. The blue bars indicate the relative AEP loss for the single turbine and the red bars show the relative AEP loss for the full wind farm. For both cases, the mean AEP loss over the full simulation is listed in the legend. Based on 10 years of operation, we observe a difference in AEP loss between simulating a single wind turbine compared to an entire wind farm. Generally, the trends from the single turbine and the wind farm are very similar. The AEP loss is observed to be relatively small for the first 3-4 years but quickly increases after this point. We also observe that the first 5 years of operation account for less than 15 % of the total loss for both cases. As mentioned, the main difference between the two cases is the magnitude of the power loss. For both cases, the highest AEP loss is observed to occur in the two last years of the simulation with a maximum AEP loss of 2.9 % for the single turbine and 2.7 % for the wind farm. This corresponds to a 7 % reduction from simulating the single wind turbines versus the full wind farm. Since the power loss is relative to its non-eroded counterpart, the wind farm case will naturally have a smaller loss because of the added wake losses, which reduce the overall energy production. This can also be verified from the per-turbine AEP which was found to be 9.7 GWh for the single wind turbine but only 8.9 GWh for the entire wind farm. The average AEP loss for the entire simulation period was 1.51 % for the single turbine and 1.40 % for the wind farm, corresponding to an 7 % reduction.



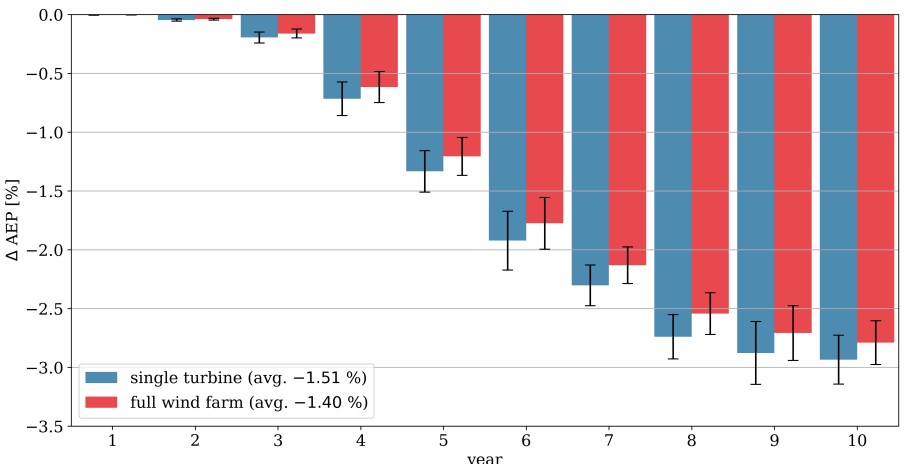

**Figure 7.** Comparison of relative AEP change for a single turbine vs. the full wind farm. In both cases, the AEP is calculated relative to the non-eroded counterpart. The graphs shows the simulation over a period of 10 years using 10 seeds to compute mean and standard deviations.

270 Since the simulations is run for a period of 10 years, it is possible to assess the aerodynamic condition of the blade at certain points throughout the simulation. Figure 8 shows snapshots of the wind farm-average lift-over-drag loss along the normalized blade after year 1, 3, 5 and 10. In all cases, the maximum loss is assumed to occur at the tip of the blade from where it decreases towards the root. It is also observed that the aerodynamic loss is only affecting a certain extent of the blade. We observe that the distribution of aerodynamic loss is not changing linearly along the blade and also not changing linearly over time.

275 After year 1, the erosion damage is very small and barely causes any aerodynamic losses. After year 3, the maximum loss at the tip is around 3 % and decreases smoothly inwards. This appearance would resemble a slightly roughened surface on the outer 20 % of the blade, that cause insignificant AEP losses. After year 5, we observe a significant increase in maximum aerodynamic loss. At the tip, the loss is on average around 15 % but decreases rapidly towards the root. This appearance resembles the initiation of more severe erosion locally at the tip. The aerodynamic loss is here observed to decrease much

280 faster when moving towards the root. After year 10, the maximum loss reaches around 45 %, which is close to the obtainable maximum. We also observe that a larger extent of the outer blade is expected to experience significant aerodynamic loss. This would resemble that the erosion at the tip is starting to stagnate in terms of aerodynamic loss. The aerodynamic loss will not increase much anymore, but the damage progression will happen in the form of a much larger damage extent. We also observe that a larger extent of the blade is affected by a roughened surface.

285 It should be mentioned, that the damage prediction model is limited by the range of the outputted encoded damage (ranges between 0 and 1). This also limits the aerodynamic loss since the two models are directly coupled. In real-life conditions, it is expected that damages beyond this range could occur if the blade is left fully untreated. However, it is also expected that blades are typically inspected annually and repaired accordingly, i.e., before extreme damage occur. As mentioned, the damage





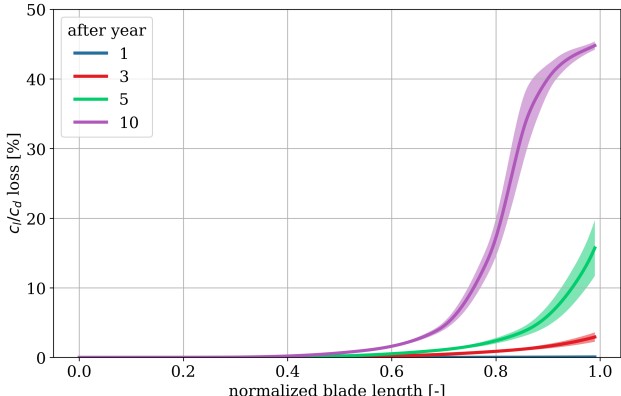

**Figure 8.** Snapshots of the wind farm-average lift-over-drag losses along the blade after 1, 3, 5 and 10 years of operation.

encoding scheme used by the damage prediction model, takes its basis in the range of allowable defects observed from real
blade inspections.

As stated earlier, the aerodynamic properties governing the wind farm simulation are the power and thrust curves. The thrust
coefficient governs the wake behavior in the engineering wake model and the power curve directly affects the energy produc-
tion. Since we make an explicit distinction between these two properties, we are able to directly separate the contributions from
each of the two. This is done by running two parallel simulations where the power and thrust coefficient curves are degraded
independently, i.e., the power curve is eroded and the thrust coefficient it assumed clean, and vice versa. This allows to exactly
quantify which contribution comes from each of the two properties.

Figure 9 visualizes the relative AEP loss from the isolated aerodynamic performance properties over the full simulation
period of 10 years. As expected, the isolated effect on the power curve is observed to cause a negative effect on the overall
AEP. It is also observed that the isolated contribution from the eroded power curve corresponds very closely to the relative AEP
loss observed for the single-turbine case in Figure 7. This indicates that erosion-induced power curve degradation does not vary
significantly, and the main variation comes from the reduced local wind speed caused by wakes. Oppositely, the isolated effect
from the eroded thrust coefficient is observed to positively contribute towards AEP. At first, it might seem counter-intuitive that
erosion can actually contribute to a power gain. However, we are assuming that erosion reduces the aerodynamic efficiency,
which also reduces the thrust force which the rotor exerts on the wind, thereby diminishing the wake deficit and leaving more
energy for downstream turbines. The contribution is solely relevant for wind farms where wake losses are present. Combining
the individual contributions from power and thrust coefficient, we end up with an overall loss, corresponding exactly to the
relative power change observed in Figure 7.

Until now, we have focused on AEP, which provide key information in a simple manner. Since the simulations are based on
half-hourly data we can also analyze the results directly at this temporal frequency. Figure 10 shows the pairwise relationships
between half-hourly variables from the full wind farm simulation. The row of sub-figures shows the relative power loss plotted
against three other variables, namely the wind speed (left), wind direction (middle) and wake loss (right). The plots are based



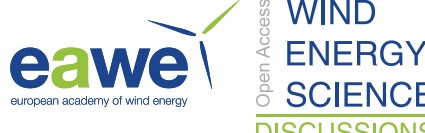

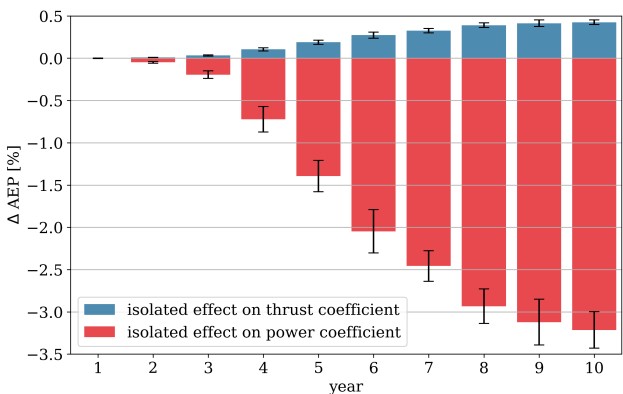

**Figure 9.** Relative AEP changes from isolated aerodynamic rotor properties. The red bars indicate the isolated contribution coming from the power coefficient and the blue bars indicate the isolated contribution coming from the thrust coefficient.

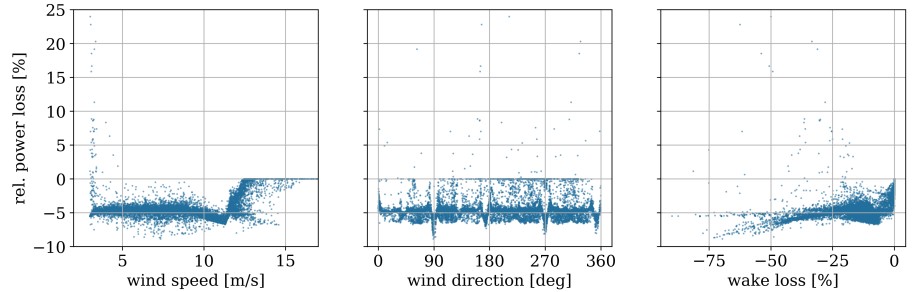

**Figure 10.** Correlation plot showing the relation between the power loss and free stream wind speed, wind direction, and reference wake loss.

on data only from the last year (year 10) where the effects from LEE are most apparent. Firstly, we identify that the majority of the simulated hours result in power loss with losses up to 7 % and that these occur mostly in the operational region below rated. For wind speeds below 10 m/s, the relative power loss is centered around -5 %, though with a lot of scatter. The loss
diminishes for wind speeds above rated where the effect of LEE is also expected to be gone. Several observations have a high erosion-induced power gain of up to 25 %. However, looking at the relation between relative power and wind speed, we identify these instances only occur at low wind speeds (below 5 m/s), thereby having a limited impact on the overall power loss. During these unique hours, the effect from reduced wake deficits overtake the effect from the degraded power curve, leading to an overall power gain.Wake losses are heavily dependent on wind direction and it is expected that erosion-induced wake
loss mitigation is also wind direction dependent. This is apparent when looking at the relation between power loss and wind direction. Certain periodic wind directions cause a consistently smaller or higher relative power loss than others. These wind directions correspond exactly with the physical alignment of the turbine rows, i.e., along the rows of the wind farm layout. The relation between power loss and wake loss supports that the highest relative power loss occurs for highly wakes instances.



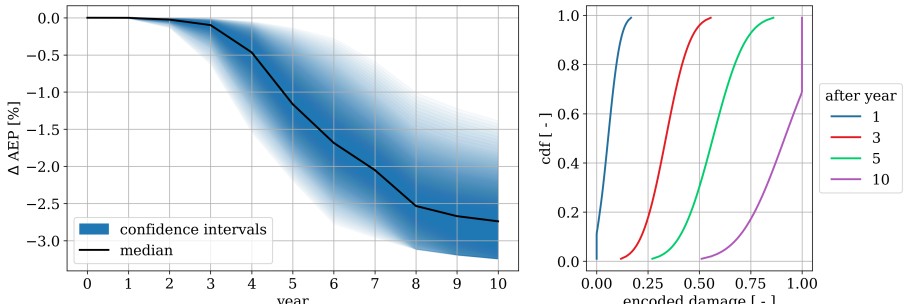

**Figure 11. (left)** Confidence intervals of the relative wind farm AEP loss with the median indicated by the black line and **(right)** the cumulative density functions of the encoded damage after years 1, 3, 5, and 10.

Finally, it should be mentioned that we have only considered the relative power loss. It was found that 95 % of the absolute power loss occurred in wind speed region between 8-14 m/s.

## 4.2 Probabilistic simulations

Previous simulations have been performed using the ensemble mean as the sole deterministic output from the damage prediction model. In this case, the only damage variation across the wind farm comes from the wake-induced changes in effective wind speed observed by each wind turbine. This variation effectively changes the operational rotor speed of the individual turbines, but the maximum percentage difference for the aerodynamic losses across the wind farm was found to be less than 1 %.

The ensemble capabilities of the damage prediction model also allow for making probabilistic damage estimates, thereby introducing the uncertainty captured by the damage model. Initially, we will use the ensemble inverse cumulative density function (CDF) to provide a probabilistic damage output. Figure 11 shows the probabilistic simulation results for the full wind farm using the same 10 years of weather data as inputs. The left figure shows the 95 % confidence interval of the relative AEP loss with the 50$^{th}$ percentile (median) indicated by the solid line. The right figure shows the CDF of the encoded damage at years 1, 3, 5, and 10.

From the relative AEP loss, we observe the uncertainty being propagated from the damage prediction model. The model uncertainty is very low for the first years but rapidly increases after year 2. At the 10th year, we expect the relative AEP loss to range between 1.4-3.2 % based on the 95 % CI. We observe a slightly asymmetric model uncertainty with higher confidence weight towards the upper quantiles, indicated by the CI bands being slightly more narrow. It is especially visible for the last couple of years. This is also observed from the cumulative density function which shows a slight asymmetry around the median, that is becoming profound throughout the years. The output range of the damage prediction model is also clearly visible from the cumulative density function, with a large proportion of probability being constrained at year 10. Finally, we observe the feature of the incubation period which also captured by the damage prediction model.

In addition to directly outputting a specific damage percentile for all wind turbines, the damage across the wind farm can be randomly sampled using the inverse cumulative density function, i.e., using Monte Carlo (MC) sampling. This would better





represent the stochastic damage distribution observed from field inspections. Based on 250 random MC simulations, the AEP variability was found to increase over time, but the maximum percentage difference (between most/least productive) after 10 years was found to be less than 0.2 %. Of course, this should be seen in the light of the total energy production of the 350 entire wind farm and the corresponding revenue might end up being considerable. Since we are simulating the full wind farm for each random instance, the aggregated energy production of the 80 wind turbines ensures convergence of the summary statistics, which is why the variability is so low. If the same experiment was performed for individual wind turbines or a much smaller wind farm, the variability would be expected to be correspondingly larger.

Finally, the random sampling provides an indication of the modeling sensitivity which could potentially be used to better 355 prioritize repairs. For each MC simulation, we will assume that one turbine is fully repaired after year 9, i.e., its aerodynamic properties are reset to a clean state. We iterate through every turbine in the wind farm and evaluate the relative impact on the AEP for the remaining year, i.e. year 10. This procedure is performed for all MC simulations and allows for prioritizing the order in which each turbine should be repaired to gain the maximum energy production. For each MC simulation we get 80 samples, one for each wind turbine repair, which can be used to assess the effect from that individual turbine. Due to 360 computational costs, a simpler wake model was used where only the steady wake deficits are considered, i.e., no turbulence model is included. The results are analyzed quantitatively by ranking them according to their individual AEP gain. The results are summarized in Figure 12 which shows, on the left-side, the direct relation between the AEP gain and the damage level of the repaired turbine. Each point represent a repaired turbine and the total number of points is therefore $N_{MC} \cdot N_{wts} = 250 \cdot 80 = 20,000$. In addition to showing the relation between AEP gain and damage level, the points are also colored according to their 365 reference AEP contribution which is shown on the right-side plot. Here, we show the mean AEP contribution from each of the turbines in the wind farm based on the 250 MC simulations. Not surprisingly, the inner turbines are expected to contribute less since they will more frequently be operating in the wake deficit of the outer turbines. The biggest contributor is the turbine located in the south-western corner, which is contributing almost 9% more than the smallest contributor.

It is seen that a very strong correlation is found between the encoded erosion damage and the added energy production. 370 This verifies that, generally, the overall biggest gain in energy production is obtained by repairing the most damaged turbine. This also corresponds with the previous findings, showing that the largest contribution to the power loss comes from the eroded power curve. Secondly, we also observe a correlation between the reference AEP contribution and the added energy production. This indicates that in a case with two equally damaged wind turbines, the turbine providing the highest contribution relative to the overall energy production should be prioritized over smaller contributors. This favors the turbines positioned in the outer 375 rows of the wind farm, and especially on the side of the prevailing wind direction. We even observe several instances, where it is more favorable to repair less damaged turbines, since their relative AEP contribution is larger. Potentially, a third priority would be repairing the turbines that causes the biggest reduction in wake loss. As previously shown, LEE reduces the wake deficits which in turn contributes positively to the overall energy production. A very weak correlation was found be between wake reduction and added energy production. This indicates that an energy production gain can be achieved by prioritizing 380 turbines that contribute least to wake reduction. It is difficult to conclude if the correlation implies a causal relationship, and the prioritization would only be relevant in extremely rare instances.





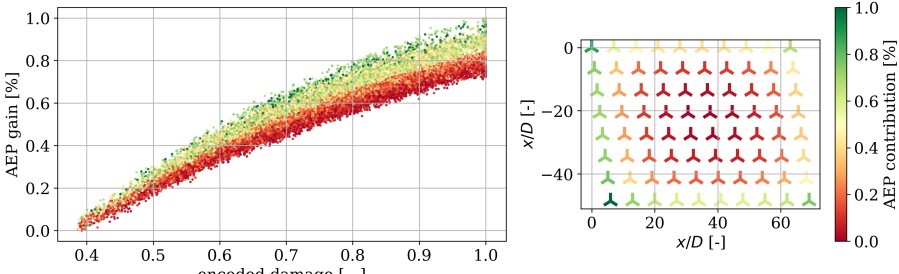

**Figure 12.** AEP gain plotted against the damage level of the repaired turbine. The colors indicate the reference AEP contribution which is also shown on the right-side figure. Both the AEP gain and reference AEP has been linearly normalized.

## 5 Discussion

The simulation of LEE is challenging considering its multidisciplinary nature, which involves several fields of science and engineering, such as aerodynamics, materials science, mechanical engineering, meteorology, etc. In the present study, we have

coupled a damage prediction model with an aerodynamic loss model, to simulate the progression and aerodynamic effect of LEE in wind farms. Many underlying assumptions affect the results of the present study, and we will try to justify, evaluate and discuss these in the following section.

Wind farm flow modeling on its own is a very complex discipline. We typically distinguish between steady-state and dy-

namic modeling, and in the present study, we have employed a steady-state Gaussian wake deficit model for estimating wake losses across the wind farm. The main assumption is the constant wind flow across the wind farm where the steady flow is assumed to be valid for the entire time bin. Since it assumes a constant wind flow, steady-state flow modeling cannot capture the effects of turbulence and other dynamic flow phenomena that occur in the atmosphere. E.g. it cannot account for any transient effects such as sudden changes in wind speed or direction, change in atmospheric stability, and other meteorological

phenomena.

On the other hand, dynamic wind farm flow modeling is a more advanced technique that takes into account the dynamic nature of the wind flow. It considers changes in wind speed, wind direction, and turbulence at a much higher temporal resolution. The simulation includes detailed information about wind turbine interactions and the surrounding atmosphere. Dynamic modeling is more complex and computationally expensive compared to steady-state modeling, but it does provide more accurate

and detailed information about wind farm performance. In addition, dynamic wind farm flow modeling requires a much more detailed and accurate representation of wind turbines and the surrounding environment, which can be challenging to obtain.

The scope of the present study was to demonstrate the aerodynamic effects of LEE in wind farms through a combination of erosion, aerodynamic, and wake modeling. Since the requirements for dynamically modeling all of these properties over long time periods were unfeasible, the simulations were performed using steady-state wind farm flow modeling. This type of

modeling fidelity is commonly used for long term simulation of wind farms where the low-order statistical moments are of





interest. However, a future study investigating high-fidelity flow modeling of eroded wind turbines is required to accurately assess the dynamics as well. Such an analysis could be carried out for specific atmospheric and meteorological conditions as well predefined erosion levels, to limit the computational requirements. It could similarly be used with higher fidelity wind turbine


One of the main uncertainties related to erosion modeling is the assumed relationship between the output of the damage prediction model which is used as input for the aerodynamic loss model. As previously described, it is very difficult converting between structural and aerodynamic defects as they are not weighted equally. Current repair recommendations are primarily based on structural integrity but there might be a potential financial benefit from also considering the aerodynamic impact.
The conversion table from Figure 3 provides the relationship used for the study and is based on results from a collection of literature and CFD simulations. However, it could be of interest to propose an additional uncertainty parameter to this, currently, deterministic relationship. This would allow for mimicking the uncertainty related defect categorization from blade inspections.

In the present study, the output from the damage prediction model was always taken to be located at the tip and decreasing
with a cubic relation towards the root. It was shown by Visbech et al. (2023), that the average distribution of defects from more than 550 blade inspections were following a similar profile. For that reason, we argue that the statistical distribution is more realistically represented by a continuous function rather than a step function as usually suggested in the literature (Gaudern, 2014; Sareen et al., 2014; Maniaci et al., 2020). However, blade inspections and laboratory testing of newer coating materials show that erosion defects tend to be more randomly distributed along the blade and do not always initiate at the tip. For that
reason, it could be of interest to implement a stochastic defect distribution which would better resemble this behavior.

One of the key challenges related to modeling aerodynamic loss from LEE is validating the results. Several assumptions were made for the aerodynamic loss model used in the present study, in order to make the modeling process more tractable. These assumptions can introduce uncertainties, particularly if the assumptions are not well-validated or do not accurately reflect real-world conditions. Validating the aerodynamic degradation of power curves is incredibly difficult using operational
data such as that obtained from the SCADA system. For that reason, the main type of validation will still have to come from high-fidelity aeroelastic simulation tools. Only recently was the aerodynamic effect of LEE quantified using operational field data (Panthi and Iungo, 2023).

LEE has been expected to have an even larger impact on wind turbines in the future (Pryor et al., 2021; Shields et al., 2021).
Generally, we observe new turbines to have longer blades, operating at higher tip speeds, which effectively increases the rate at which erosion initiates and develops. At the same time, new leading edge protection (LEP) systems are being developed to better withstand the more severe operational conditions. These effects contravene and complicate the generalizability of LEE modeling in the future.

We also expect wind farms to become larger in size, which would change the wind farm flow field. In the present study,
the effect on the power curve would not be expected to change considerably since the damage prediction model is not very





sensitive to changes in tip speed (why we also see very little inter-turbine damage variability for the wind farm). However, the effects from eroded thrust coefficient would scale non-linearly with wake loss. Therefore, the potential wake reduction is dependent on the layout.

Site-specific weather conditions might also influence the results presented in this study. Less windy sites would result in
a longer incubation periods but since LEE only affects the aerodynamic properties below rated, it would lead to a relatively larger power loss. Other sites might have a strong correlation between rain and wind direction (e.g., close to a mountain ridge), which would lead to more damage variation across the wind farm. Adding uncertainty properties to the weather inputs (wind speed, wind direction and rain), could allow for better addressing the sensitivity to these parameters.

LEE has started to be considered as a potential operation and maintenance improvement in wind farm control (Meyers et al., 2022), e.g., through erosion-safe operation as demonstrated by Bech et al. (2018). With this mitigating strategy, the rain impinging the blade is reduced through rotor speed reductions during episodes of heavy rain. The damage is thereby reduced at the cost of energy production. Erosion-safe operation has however not been demonstrated in real life, nor has it been implemented within wind farm flow control. Implementing erosion-safe operation in farm control strategies would require the modeling of
wake impacts between the turbines and estimating the AEP loss due to eroded blades in the wind farm. To work towards a wind farm flow control algorithm that includes erosion-safe operation, we first need a modeling framework that can predict damage progression and AEP loss within a wind farm.

Finally, it can be mentioned that the modeling framework presented in this paper could potentially be coupled with real blade
inspections following a Bayesian updating scheme approach (Asgarpour and Sørensen, 2018). The blade inspections would provide true damage distributions for all blades, thereby diminishing the model uncertainty at the time of inspection. The damage prediction model could be calibrated to match the observed damage distribution, and the associated past AEP loss would then be obtained through propagating back in time. Having calibrated the damage prediction model, it could be used to estimate expected AEP loss for a given period assuming no repair, which would allow for much more informed decision-making.

**6 Conclusions**

LEE affects the aerodynamic properties of a wind turbine and thereby reduces rotor performance. This directly decreases the energy production but it potentially also reduces the overall wake losses, which is not considered when modeling LEE on individual turbines.

In this study, we have used a modeling framework that combines damage prediction and aerodynamic loss modeling with
steady-state wind farm flow modeling. The framework can be used to efficiently simulate the long-term aerodynamic effects of LEE using site-specific mesoscale weather data and basic wind farm specifications.

The modeling framework was used to simulate the development of LEE on an offshore wind farm over a simulation period of 10 years. Comparing the wind farm simulation to that of a single wind turbine, it was found that AEP loss were overestimated





with up to 7 % when neglecting the effect on wake reduction. The average AEP loss for the wind farm was found to 1.4 % with
a maximum AEP loss of 2.7 % occurring at the last year.

A Monte-Carlo-based sensitivity analysis was carried out to establish a probabilistic priority list of turbines which should be repaired first to maximize energy production. It was found that the level of erosion damage was generally the governing factor, but specific turbines which contribute relatively more to the overall energy production, should be prioritized in certain cases to maximize energy production.

The main limitations of the study are related to the coupling between structural and aerodynamic damages and verification of the aerodynamic losses through simulations of higher fidelity. Future work should emphasize on uncertainty quantification through probabilistic modeling which could be coupled to real inspection¨data through a Bayesian updating scheme.

*Code and data availability.* Software code developed and used for this study will be made publicly available. Parts of the data can also be made available upon request from the corresponding author (Jens Visbech).

**Appendix A: PyWake setup specifications**

**Table A1.** List of the engineering models used in the PyWake simulation setup.

| Models | Name | Superposition |
|---|---|---|
| Wake deficit model | NiayifarGaussianDeficit | linear sum |
| Turbulence model | CrespoHernandez | max. sum |
| Blockage model | None | - |
| Rotor average model | RotorCenter | - |
| Ground model | None | - |
| Deflection model | None | - |

*Author contributions.* JV had the lead on paper writing and software development. TG contributed mainly to the damage prediction and wind farm flow modeling. PER contributed mainly to the wind farm flow modeling and PyWake implementation. OSO and AMF contributed mainly to the aerodynamic loss model. AMF devised the aerodynamic loss categories and conversion scheme. AH contributed mainly to the design of the overall project. All authors contributed to writing and editing the paper.

*Competing interests.* The authors declare that they have no conflict of interest.



*Acknowledgements.* The authors would like to acknowledge everyone at DTU Wind and Energy Systems who was involved in the LEE Cross-Cutting Activity (CCA) which was carried out through 2022.



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
