# Peer review of "Aerodynamic effects of leading edge erosion in wind farm flow modeling"

_Wind Energy Science, 2023_

## Referee Comment (RC1)

**_General comments:_**
"Aerodynamic effects of leading edge erosion in wind farm flow modeling" studies the effect of leading edge erosion (LEE) on wind farm energy production directly through degraded power and thrust curves including the effect on wake losses. It is argued that eroded blades will decrease the energy production of an individual turbine, but at the same time, the wake deficit will also be reduced leading to a wake containing a higher energy for the downstream turbines. To this end an existing damage prediction model is coupled with a proposed aerodynamic loss prediction method to form the combined LEE module. The numerical simulations were performed using a steady wind farm flow model. LEE induced power losses are compared between an individual turbine and an entire wind farm. Finally, a prioritized repair strategy is propose based on Monte Carlo simulations.

The main focus of the study is based on determining the LEE damage of an individual turbine for a given time series of weather inputs (wind speed and rain) and turbine operational data of a site. An existing damage prediction model is combined with a newly proposed aerodynamic damage prediction model to determine the blade-sectional aerodynamic losses. The eroded power and thrust curves are obtained based on these sectional losses are used for a numerical simulation of a wind farm. The input of these numerical simulations are a time series of wind speed, wind direction, and rain to simulate the gradual development of erosion on each individual turbine in a wind farm with a possibility of updating the modeled damage state at a given time step.

The first part of the LEE module is an existing structural erosion damage prediction model that provides estimate of the erosion damage along the blade, based on time series of turbine-local wind speed and rain. In the second part of the LEE module an aerodynamic losses categorization is introduced based on the erosion level and Reynolds number. Finally, these two models are combined to obtain the LEE module.

The aerodynamic loss categorization and combination with the structural damage categorization needs further explanations. Please see below for specific comments.

**_Specific comments:_**
1. **pages 5-6, section 2.3:** In this section the aerodynamic loss model is very briefly discussed. There are several unclear points:

   1.1. **lines 133-134:** The manuscript reads: "_Indeed, the maximum loss registered for erosion-type damages never exceeded 64% and once roughness caused the transition to occur at the leading edge, the additional loss from more severe erosion was between 10 − 15%._"

   It is stated that the maximum loss is 64%, but also argued about additional 10-15% loss. These statements are not clear. Please elaborate what is that additional severe erosion and how can a damage never exceed 64% but can have additional 10-15% losses?

   1.2. **lines 138-142:** The manuscript reads: "_To assess the aerodynamic losses at higher Reynolds numbers 2D CFD computations were performed for four airfoils with relative thickness below 21 % (NACA63-418, FFA-W3-211, DU96-W-180, Risø-B1-18, Risø-C2-18) for a Reynolds number range from 5 − 15 million and different levels of erosion. The latter was generated using the spectral approach detailed in (Meyer Forsting et al., 2022a) and combined with a forward-facing step (height of 1.5 × 10−3 in chord units) on the suction side to represent the worst erosion level. The surface perturbations were directly resolved in those simulations._"

   The statement suggests that there were more Reynolds numbers considered, but in the results (e.g.: table in figure 3b), only 2 Re numbers are mentioned. Please clearly indicate if more Reynolds numbers are considered or not.

   - What are these various erosion levels and how are they applied in CFD simulations?
   - What is meant by "the latter was generated using a spectral approach"?
   - What is exactly simulated with "spectral approach"?
   - Were there more CFD methods applied? If so, what other methods are used?

   1.3. It seems that the worst erosion level is modeled by a forward-facing step:

   - How does this forward-facing step look like?
   - How was the airfoil geometry modified to obtain this forward-facing step?
   - At which chord location was this step applied?

- What are the mentioned surface perturbations?

Please provide a detailed explanation of the modeling of the erosion in the CFD simulations. Please also provide a brief description of the CFD methods employed and the basic set-up of the cases (e.g. number of grid elements, domain sizes, boundary conditions, etc.) for the sake of the repeatability and comparison for the possible future studies.

1.4. **lines 144-145:** What is meant by "*diminishing drop by about 15 percentage*"? Could you please indicate lift-to-drag ratios for both Re numbers of 5e6 and 15e6 and re-formulate this sentence?

1.5. **lines 145-147:** It is not clear how these 5 categories were defined and what do these categories refer to. Are these categories based on the erosion level/height? Also, what are the values represented in table 3b?

1.6. **lines 147-148:** The manuscript reads: "*The first two categories capture the losses associated with the gradual movement of the transition location towards the leading edge over the entire blade section in question.*"

- Are categories "a" and "b" the first 2 or "b" and "c"?
- It is not clear how these categories capture the losses. Please explain how/why the transition location is moving towards the leading edge.
- Does the category "a" refer to a clean (no erosion) case? Please mention that.

1.7. **line 149**: It is stated that pits and gouges are assumed. Do you mean that one of these categories (is category c meant here) can be related to pits and gouges of the real wind turbine blade situation? If this is the case, how are other categories related to the real erosion cases?

1.8. **lines 149-150:** It is not clear how the additional losses are caused. It is also not clear what is meant by surface roughness and sharp steps. Is the surface roughness mentioned here different than pits and gouges? Are these sharp steps caused by erosion? Do you mean to relate the categories with real cases in these statements?

1.9. **line 152:** Although a reference is given, I suggest to briefly explain how a 60:40 mix of transitional and fully turbulent performance can be obtained/calculated.

1.10. **lines 153-154:** What kind of losses are these? Are these the drop in lift-to-drag ratio (in percent)?

**Section 2.3 – general remark:** Please explain the CFD simulations set up briefly and explain the procedure of defining the aerodynamic loss categorization in detail. It should be possible to generate this categorization for different Reynolds numbers based on this study.

2. **Section 2.4:** It is not explained how the structural damage prediction model is coupled with the aerodynamic loss categorization to obtain figure 3c. Please provide details of this "summing" procedure.

3. **lines 192-193:** Could you please present the mentioned empirical relation and explain how the losses included into this relation?

4. **Figure 5 caption and line 240:** 2 different values are given for mean annual rainfall. Please check the values.

5. **line 264:** "This corresponds to a 7% reduction..": is this a reduction in losses?

6. **lines 264-269:** Here it is argued that when the single turbine is considered the AEP is higher although the AEP loss is also higher compared to the wind farm case. This discussion needs a further elaboration/clearance.

7. **Figure 8:** It is not clear how exactly this graph is obtained. Maybe if the tables/graph in figure 3 explained in detail then this graph can be understood better. What is the (range of) Reynolds number considered for each blade section? Did you inter/extrapolate the values given in figure 3c for

different Reynolds numbers?

8. **line 323:** Could you please explain what is meant by "... highly wakes instances"?

9. **lines 343-344:** It is stated that: "It is especially visible for the last couple of years." Could you please be more specific?

10. **lines 343-344:** What is an incubation period for this case and how this "feature" is observed?

11. **section 5:** Very detailed and clear discussion on the applicability, shortcomings, and improvement points of the current model. Thanks for including this discussion here.

***Technical corrections:***
**line 165**: "between" is repeated twice.
**Figures 3 a, b and c:** Please enlarge these figures for better readability.
**Figure 3b:** please indicate the unit of the presented values.
**Table 1:** Is this table really necessary? Isn't it possible to mention these properties within the text?
**line 269:** "an 7% reduction" should be "a .."
**line 270:** ".. is run.." should be ".. are run .."
**line 319:** ".. gain.Wake .." there should be a space ".. gain. Wake.."
**line 378:** ".. was found be between .." should be ".. was found to be between .."
**line 474:** ".. was found to 1.4 % .." should be ".. was found to be 1.4 % .."

---

## Author Comment (AC1)

**Response to Anonymous Referee #1 comments of Manuscript ID WES-2023-128 entitled "Aerodynamic effects of leading edge erosion in wind farm flow modeling"**

Thank you for taking the time to review our article. We have addressed your comments attentively, for which the details are provided below.

**Specific comments**

:

1. pages 5-6, section 2.3: In this section the aerodynamic loss model is very briefly discussed. There are several unclear points:

   (a) lines 133-134: The manuscript reads: "Indeed, the maximum loss registered for erosion-type damages never exceeded 64% and once roughness caused the transition to occur at the leading edge, the additional loss from more severe erosion was between 10-15%."
   It is stated that the maximum loss is 64%, but also argued about additional 10-15% loss. These statements are not clear. Please elaborate what is that additional severe erosion and how can a damage never exceed 64% but can have additional 10-15% losses?
   The sentence can easily lead to confusion. Transition moving to the leading edge leads to a drop of up to 50 % and then severe erosion can add another 15 %, so you arrive at around 65 %. The sentence was simplified.

   (b) lines 138-142: The manuscript reads: "To assess the aerodynamic losses at higher Reynolds numbers 2D CFD computations were performed for four airfoils with relative thickness below 21 % (NACA63-418, FFA-W3-211, DU96-W-180, Risø-B1-18, Risø-C2-18) for a Reynolds number range from 5 - 15 million and different levels of erosion. The latter was generated using the spectral approach detailed in (Meyer Forsting et al., 2022a) and combined with a forward-facing step (height of $1.5 \times 10^{-3}$ in chord units) on the suction side to represent the worst erosion level. The surface perturbations were directly resolved in those simulations."
   The statement suggests that there were more Reynolds numbers considered, but in the results (e.g.: table in figure 3b), only 2 Re numbers are mentioned. Please clearly indicate if more Reynolds numbers are considered or not.
   - What are these various erosion levels and how are they applied in CFD simulations?
   - What is meant by "the latter was generated using a spectral approach"?
   - What is exactly simulated with "spectral approach"?
   - Were there more CFD methods applied? If so, what other methods are used?

   Please provide a detailed explanation of the modeling of the erosion in the CFD simulations. Please also provide a brief description of the CFD methods employed and the basic set-up of the cases (e.g. number of grid elements, domain sizes, boundary conditions, etc.) for the sake of the repeatability and comparison for the possible future studies.
   The number of Reynolds numbers are now mentioned explicitly in the text, but losses are only reported at both extremes. As the categories are already approximations, linearly interpolating between them is deemed sufficiently accurate. Furthermore we have added how the spectral roughness was generated and how the CFD simulations were performed, including the numerical setup. For details we refer to two papers, which should allow repeating our simulations, especially as the surface grid generator is open-source.

   (c) lines 144-145: What is meant by "diminishing drop by about 15 percentage"? Could you please indicate lift-to-drag ratios for both Re numbers of 5e6 and 15e6 and re-formulate this sentence?
   This sentence was removed and the entire paragraph rewritten to give more detail on how the categories were conceived. Some sentences were modified and moved. We know also clearly state that the categorization is a first attempt at trying to categorize leading edge damages aerodynamically.

   (d) lines 145-147: It is not clear how these 5 categories were defined and what do these categories refer to. Are these categories based on the erosion level/height? Also, what are the values represented in table 3b?

It is related to the type of damage observed on the blade. This is now more clearly defined in the text. Shown are losses in lift-to-drag, which should now be more easily understood after rewriting this section.

    (e) lines 147-148: The manuscript reads: "The first two categories capture the losses associated with the gradual movement of the transition location towards the leading edge over the entire blade section in question."

- Are categories "a" and "b" the first 2 or "b" and "c"?
- It is not clear how these categories capture the losses. Please explain how/why the transition location is moving towards the leading edge.
- Does the category "a" refer to a clean (no erosion) case? Please mention that.

Yes, category "a" refers to a clean blade. This has been clarified in the manuscript. The section was rewritten and now clearly states what the categories correspond to visually.

    (f) line 149: It is stated that pits and gouges are assumed. Do you mean that one of these categories (is category c meant here) can be related to pits and gouges of the real wind turbine blade situation? If this is the case, how are other categories related to the real erosion cases?
Hopefully the new text clarifies this, but of course the categories are uncertain and might be revised in the future.

    (g) lines 149-150: It is not clear how the additional losses are caused. It is also not clear what is meant by surface roughness and sharp steps. Is the surface roughness mentioned here different than pits and gouges? Are these sharp steps caused by erosion? Do you mean to relate the categories with real cases in these statements?
Yes we hope that we can have some equivalence to what is seen in inspection images, but of course this is difficult and subject to uncertainties.

    (h) line 152: Although a reference is given, I suggest to briefly explain how a 60:40 mix of transitional and fully turbulent performance can be obtained/calculated.
We added a footnote.

    (i) lines 153-154: What kind of losses are these? Are these the drop in lift-to-drag ratio (in percent)?
Yes this is now clarified.

2. general remark: Please explain the CFD simulations set up briefly and explain the procedure of defining the aerodynamic loss categorization in detail. It should be possible to generate this categorization for different Reynolds numbers based on this study.
We appreciate the detailed comments and have tried to address them by rewriting most of the section and adding information where it was lacking. As it is now also mentioned in the text, it is a first attempt at defining aerodynamic categories and more work remains to be done here.

3. It is not explained how the structural damage prediction model is coupled with the aerodynamic loss categorization to obtain figure 3c. Please provide details of this "summing" procedure.
The two individual schemes are not summed, and the authors agree that this is confusing due to the "+" symbol between figure 3(a) and 3(b) which has now been removed. Figure 3(a) and 3(b) show how each of the two submodels (damage prediction model and aerodynamic loss model) defines erosion categories based on their respective papers (see Visbech, Jens, et al. "Introducing a data-driven approach to predict site-specific leading-edge erosion from mesoscale weather simulations", Wind Energ. Sci., 8, 2023 and Bak, Christian. "A simple model to predict the energy loss due to leading edge roughness." Journal of Physics: Conference Series. Vol. 2265. No. 3. IOP Publishing, 2022). These two definitions have to be combined, and figure 3(c) shows how we relate the encoded damage, shown on the x-axis, to the aerodynamic losses (percentage loss in lift-to-drag ratio), shown on the y-axis. This has also now been clarified in the manuscript.

4. lines 192-193: Could you please present the mentioned empirical relation and explain how the losses included into this relation?
The empirical relation was obtained from the paper introducing the aerodynamic loss model (see Bak, Christian. "A simple model to predict the energy loss due to leading edge roughness." Journal of Physics: Conference Series. Vol. 2265. No. 3. IOP Publishing, 2022), more specifically Eq. 17. The aerodynamic model has also been validated against a blade element momentum (BEM) model. For more details on the equations, we refer to the open-source aerodynamic loss tool (see Christian Bak, & Meyer Forsting, A. R. (2023). SALT - Simplified Aerodynamic Loss Tool (1.0.0 - beta). DTU Wind, Technical University of Denmark. https://doi.org/10.5281/zenodo.7906333)

5. Figure 5 caption and line 240: 2 different values are given for mean annual rainfall. Please check the values.

   Yes, these values should indeed be identical. This has been corrected.

6. line 264: "This corresponds to a 7% reduction..": is this a reduction in losses?

   This represents the reduction in AEP loss from modeling the full wind farm compared to the single turbine, i.e., going from 2.9 % to 2.7 %.

7. lines 264-269: Here it is argued that when the single turbine is considered the AEP is higher although the AEP loss is also higher compared to the wind farm case. This discussion needs a further elaboration/clearance.

   What is meant here, is simply that the absolute per-turbine AEP is higher for a single turbine since it is not exposed to wake effects like in a wind farm. However, by considering relative losses, i.e., to the non-eroded counterpart we are able to fairly compare the AEP losses despite this difference in absolute per-turbine AEP. The manuscript has been simplified to better clarify this.

8. Figure 8: It is not clear how exactly this graph is obtained. Maybe if the tables/graph in figure 3 explained in detail then this graph can be understood better. What is the (range of) Reynolds number considered for each blade section? Did you inter/extrapolate the values given in figure 3c for different Reynolds numbers?

   Figure 8 shows the losses in lift-to-drag ratio along the blade after different years. The damage prediction model allows for estimating the structural damage along the blade based on the rotational speed and thereby the local tangential velocity. This value is converted into an aerodynamic loss based on the relationship presented in Figure 3(c).

   With regards to Reynolds number, a constant value of 5e+6 was used for the entire blade. This decision was based on the limited airfoil information for the Vestas V80-2MW wind turbine and the typical distribution of Reynolds numbers for wind turbines of similar size. This has been clarified in the manuscript and an additional reference has been added with typical Reynolds number distributions for modern wind turbines.

9. line 323: Could you please explain what is meant by "... highly wakes instances"?

   It should have stated "waked" and not "wakes". This has been corrected.

   What is meant by "highly waked instances" is the periods where the wake loss in the wind farm is large, e.g., when the wind is coming from certain directions that align with the wind farm layout.

10. lines 343-344: It is stated that: "It is especially visible for the last couple of years." Could you please be more specific?

    The encoded damage ranges between 0 and 1, and for some of the upper quantiles the upper limit is reached which constrains the probability distribution. This is the case for the last couple of years. The manuscript has been updated to better describe this.

11. lines 343-344: What is an incubation period for this case and how this "feature" is observed?

    The incubation period refers to the initial phase of the erosion development where the effects are not observable. This has been clarified in the manuscript.

12. section 5: Very detailed and clear discussion on the applicability, shortcomings, and improvement points of the current model. Thanks for including this discussion here.

    Thank you.

**Technical corrections**

:

1. line 165: "between" is repeated twice.

   This has been corrected.

2. Figures 3 a, b and c: Please enlarge these figures for better readability.

   The figure and text have now been made larger.

3. Figure 3b: please indicate the unit of the presented values.

   This has been updated.

4. Table 1: Is this table really necessary? Isn't it possible to mention these properties within the text?

   This table has been removed.

5. line 269: "an 7% reduction" should be "a .."

   This has been corrected.

6. line 270: ".. is run.." should be ".. are run .."
   This has been corrected.

7. line 319: ".. gain.Wake .." there should be a space ".. gain. Wake.."
   This has been corrected.

8. line 378: ".. was found be between .." should be ".. was found to be between .."
   This has been corrected.

9. line 474: ".. was found to 1.4 % .." should be ".. was found to be 1.4 % .."
   This has been corrected.

---

## Author Comment (AC2)

**Response to Anonymous Referee #2 comments of Manuscript ID WES-2023-128 entitled "Aerodynamic effects of leading edge erosion in wind farm flow modeling"**

Thank you for taking the time to review our article. We have addressed your comments attentively, for which the details are provided below.

1. I found the abstract did mention some of the critical conclusions drew by this manuscript, such as i) the AEP loss was overestimated up to 7% in previous studies; ii) due to the wake loss effect, there is an optimum repairing strategy for a wind farm to maintain its high productivity.
   The abstract has been updated to also include these critical conclusions.

2. The abbreviation AEP standing for the annual energy loss was not introduced with its full name when it first occurs.
   This has been corrected.

3. The abbreviation CI (I guess, standing for confidence interval) was not introduced with its full name when it first occurs in the description of Figure 2.
   This has been corrected.

4. The description of Figure 2 mentions that the solid line represents the ensemble mean while the label in figure indicates that it is for median, which is confusing. The author should clarify.
   This has been corrected.

5. In the text describing Figure 3(a) in line 164, the categorization of aerodynamic losses is not properly referenced. Please list the specific CFD simulations and reviews that the authors are referring to.
   As mentioned in line 120, there exists no standardized aerodynamic loss categorization scheme. Section 2.3 (Aerodynamic loss categories) is devoted to justifying the chosen values for the lift-to-drag losses based on the references provided in the section. These values are not universal since they might vary for different airfoils but it is the authors belief that the chosen values represents realistic lift-to-drag losses on the wind turbine used in the present study. We have tried to address your comment by rewriting most of the section and adding information where it was lacking. As it is now also mentioned in the text, it is a first attempt at defining aerodynamic categories and more work remains to be done here.

6. In line 165, the "between" is duplicated.
   This has been corrected.

7. I found that Table 1 is not giving much information, should the authors consider removing it?
   The authors agree that the table might be a bit redundant and it has therefore been removed.

8. The description of Figure 4 is unclear, should the "securities" be replaced by "severities"?
   Yes, it should have stated "severities" and not "securities". This has been updated.

9. I found that the Figure 4 is confusing because it is trying to deliver two things: 1) from the root to the tip of the blade, the tendency of erosion is increasing; 2) from the root to the tip of the blade, the lift-to-drag losses are increasing. Are the increasing lift-to-drag losses are dominantly caused by the increasing degrees of erosion? The authors might need to clarify to make the figure clearer.
   Figure shows an example of the typical distribution of erosion defects along the wind turbine blade. The erosion defect are categorized by their aerodynamic impact on the lift-to-drag ratio, ranging from a-f. The aerodynamic loss categories are directly associated with a reduction in lift-to-drag ratio as specified in Figure 3(b). The figure caption has been updated to better clarify this.

10. In Figure 12 right panel, I found that the color bar is confusing (especially the unit seems to be percentage). It will be good if the authors can clarify on that.
    The unit should not have been in % since it is normalized between 0 and 1. This has been corrected.
    The main take away from the figure should be that the largest AEP gain is obtained by repairing the most severely damaged turbines but that there is also an added, and clearly visible benefit from repairing the turbines that contributes more to the overall AEP, i.e., the turbines less affected by the wakes. The figure caption has been updated to clarify this.